# R-Transformer: Recurrent Neural Network Enhanced Transformer

## Abstract

Recurrent Neural Networks have long been the dominating choice for sequence modeling. However, it severely suffers from two issues: impotent in capturing very long-term dependencies and unable to parallelize the sequential computation procedure. Therefore, many non-recurrent sequence models that are built on convolution and attention operations have been proposed recently. Notably, models with multi-head attention such as Transformer have demonstrated extreme effectiveness in capturing long-term dependencies in a variety of sequence modeling tasks. Despite their success, however, these models lack necessary components to model local structures in sequences and heavily rely on position embeddings that have limited effects and require a considerable amount of design efforts. In this paper, we propose the R-Transformer which enjoys the advantages of both RNNs and the multi-head attention mechanism while avoids their respective drawbacks. The proposed model can effectively capture both local structures and global long-term dependencies in sequences without any use of position embeddings. We evaluate R-Transformer through extensive experiments with data from a wide range of domains and the empirical results show that R-Transformer outperforms the state-of-the-art methods by a large margin in most of the tasks. We have made the code and data publicly available [1].

## 1 Introduction

Recurrent Neural Networks (RNNs) especially its variants such as Long Short-Term Memory (LSTM) and Gated Recurrent Unit (GRU) have achieved great success in a wide range of sequence learning tasks including language modeling, speech recognition, recommendation, etc (Mikolov et al., 2010; Sundermeyer et al., 2012; Graves & Jaitly, 2014; Hinton et al., 2012; Hidasi et al., 2015). Despite their success, however, the recurrent structure is often troubled by two notorious issues. First, it easily suffers from gradient vanishing and exploding problems, which largely limits their ability to learn very long-term dependencies (Pascanu et al., 2013). Second, the sequential nature of both forward and backward passes makes it extremely difficult, if not impossible, to parallelize the computation, which dramatically increases the time complexity in both training and testing procedure. Therefore, many recently developed sequence learning models have completely jettisoned the recurrent structure and only rely on convolution operation or attention mechanism that are easy to parallelize and allow the information flow at an arbitrary length. Two representative models that have drawn great attention are Temporal Convolution Networks(TCN) (Bai et al., 2018) and Transformer (Vaswani et al., 2017). In a variety of sequence learning tasks, they have demonstrated comparable or even better performance than that of RNNs (Gehring et al., 2017; Bai et al., 2018; Devlin et al., 2018).

The remarkable performance achieved by such models largely comes from their ability to capture long-term dependencies in sequences. In particular, the multi-head attention mechanism in Transformer allows every position to be directly connected to any other positions in a sequence. Thus, the information can flow across positions without any intermediate loss. Nevertheless, there are two issues that can harm the effectiveness of multi-head attention mechanism for sequence learning. The first comes from the loss of sequential information of positions as it treats every position identically. To mitigate this problem, Transformer introduces position embeddings, whose effects,

---

[1]https://www.dropbox.com/sh/u35qgqnmjpywcqn/AAAITcId7DRPOD9KRooQW7i2a?dl=0

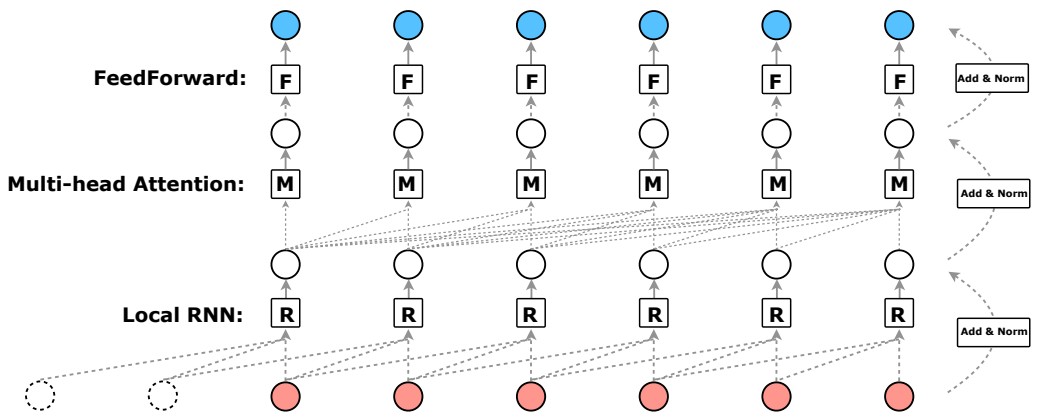

Figure 1: The illustration of one layer of R-Transformer. There are three different networks that are arranged hierarchically. In particular, the lower-level is localRNNs that process positions in a local window sequentially (This figure shows an example of local window of size 3); The middle-level is multi-head attention networks which capture the global long-term dependencies; The upper-level is Position-wise feedforward networks that conduct non-linear feature transformation. These three networks are connected by a residual and layer normalization operation. The circles with dash line are the paddings of the input sequence

however, have been shown to be limited (Dehghani et al., 2018; Al-Rfou et al., 2018). In addition, it requires considerable amount of efforts to design more effective position embeddings or different ways to incorporate them in the learning process (Dai et al., 2019). Second, while multi-head attention mechanism is able to learn the global dependencies, we argue that it ignores the local structures that are inherently important in sequences such as natural languages. Even with the help of position embeddings, the signals at local positions can still be very weak as the number of other positions is significantly more.

To address the aforementioned limitations of the standard Transformer, in this paper, we propose a novel sequence learning model, termed as R-Transformer. It is a multi-layer architecture built on RNNs and the standard Transformer, and enjoys the advantages of both worlds while naturally avoids their respective drawbacks. More specifically, before computing global dependencies of positions with the multi-head attention mechanism, we firstly refine the representation of each position such that the sequential and local information within its neighborhood can be compressed in the representation. To do this, we introduce a local recurrent neural network, referred to as LocalRNN, to process signals within a local window ending at a given position. In addition, the LocalRNN operates on local windows of all the positions identically and independently and produces a latent representation for each of them. In this way, the locality in the sequence is explicitly captured. In addition, as the local window is sliding along the sequence one position by one position, the global sequential information is also incorporated. More importantly, because the localRNN is only applied to local windows, the aforementioned two drawbacks of RNNs can be naturally mitigated. We evaluate the effectiveness of R-Transformer with a various of sequence learning tasks from different domains and the empirical results demonstrate that R-Transformer achieves much stronger performance than both TCN and standard Transformer as well as other state-of-the-art sequence models.

The rest of the paper is organized as follows: Section 2 discusses the sequence modeling problem we aim to solve; The proposed R-Transformer model is presented in Section 3. In Section 4, we describe the experimental details and discuss the results. The related work is briefly reviewed in Section 5. Section 6 concludes this work.

## 2    SEQUENCE MODELING PROBLEM

Before introducing the proposed R-Transformer model, we formally describe the sequence modeling problem. Given a sequence of length $N$: $x_1, x_2, \cdots, x_N$, we aim to learn a function that maps the

input sequence into a label space $\mathcal{Y}$: ($f : \mathcal{X}^N \to \mathcal{Y}$). Formally,

$$y = f(x_1, x_2, \cdots, x_N) \tag{1}$$

where $y \in \mathcal{Y}$ is the label of the input sequence. Depending on the definition of label $y$, many tasks can be formatted as the sequence modeling problem defined above. For example, in language modeling task, $x_t$ is the character/word in a textual sentence and $y$ is the character/word at next position (Mikolov et al., 2010); in session-based recommendation, $x_t$ is the user-item interaction in a session and $y$ is the future item that users will interact with (Hidasi et al., 2015); when $x_t$ is a nucleotide in a DNA sequence and $y$ is its function, this problem becomes a DNA function prediction task (Quang & Xie, 2016). Note that, in this paper, we do not consider the sequence-to-sequence learning problems. However, the proposed model can be easily extended to solve these problems and we will leave it as one future work.

## 3 THE R-TRANSFORMER MODEL

The proposed R-Transformer consists of a stack of identical layers. Each layer has 3 components that are organized hierarchically and the architecture of the layer structure is shown in Figure 1. As shown in the figure, the lower level is the local recurrent neural networks that are designed to model local structures in a sequence; the middle level is a multi-head attention that is able to capture global long-term dependencies; and the upper level is a position-wise feedforward networks which conducts a non-linear feature transformation. Next, we describe each level in detail.

### 3.1 LOCALRNN: MODELING LOCAL STRUCTURES

Sequential data such as natural language inherently exhibits strong local structures. Thus, it is desirable and necessary to design components to model such locality. In this subsection, we propose to take the advantage of RNNs to achieve this. Unlike previous works where RNNs are often applied to the whole sequence, we instead reorganize the original long sequence into many short sequences which only contain local information and are processed by a shared RNN independently and identically. In particular, we construct a local window of size $M$ for each target position such that the local window includes $M$ consecutive positions and ends at the target position. Thus, positions in each local window form a local short sequence, from which the shared RNN will learn a latent representation. In this way, the local structure information of each local region of the sequence is explicitly incorporated in the learned latent representations. We refer to the shared RNN as LocalRNN. Comparing to original RNN operation, LocalRNN only focuses on local short-term dependencies without considering any long-term dependencies. Figure 2 shows the different between original RNN and LocalRNN operations. Concretely, given the positions $x_{t-M+1}, x_{t-M+2}, \cdots, x_t$ of a local short sequence of length $M$, the LocalRNN processes them sequentially and outputs $M$ hidden states, the last of which is used as the representation of the local short sequences:

$$h_t = \text{LocalRNN}(x_{t-M+1}, x_{t-M+2}, \cdots, x_t) \tag{2}$$

where RNN denotes any RNN cell such as Vanilla RNN cell, LSTM, GRU, etc. To enable the model to process the sequence in an auto-regressive manner and take care that no future information is available when processing one position, we pad the input sequence by $(M-1)$ positions before the start of a sequence. Thus, from sequence perspective, the LocalRNN takes an input sequence and outputs a sequence of hidden representations that incorporate information of local regions:

$$h_1, h_2, \cdots, h_N = LocalRNN(x_1, x_2, \cdots, x_N) \tag{3}$$

The localRNN is analogous to 1-D Convolution Neural Networks where each local window is processed by convolution operations. However, the convolution operation completely ignores the sequential information of positions within the local window. Although the position embeddings have been proposed to mitigate this problem, a major deficiency of this approach is that the effectiveness of the position embedding could be limited; thus it requires considerable amount of extra efforts (Gehring et al., 2017). On the other hand, the LocalRNN is able to fully capture the sequential information within each window. In addition, the one-by-one sliding operation also naturally incorporates the global sequential information.

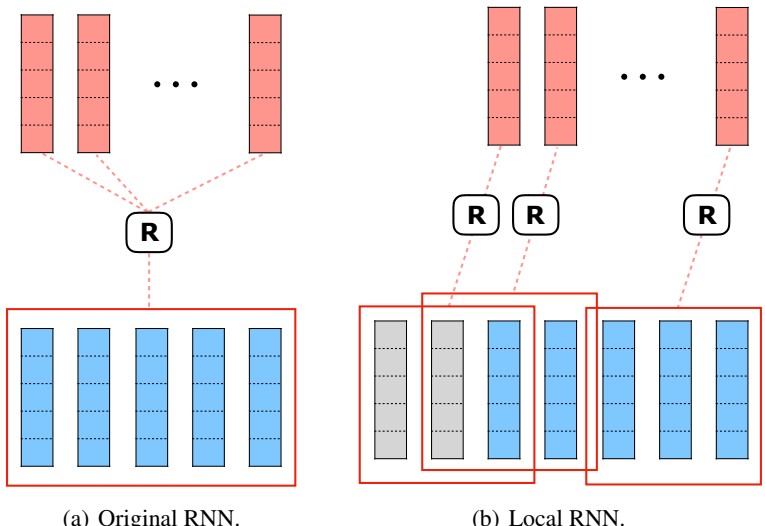

(a) Original RNN.  (b) Local RNN.

Figure 2: An illustration of the original and local RNN. In contrast to orignal RNN which maintains a hidden state at each position summarizing all the information seen so far, LocalRNN only operates on positions within a local window. At each position, LocalRNN will produce a hidden state that represents the information in the local window ending at that position.

**Discussion:** RNNs have long been a dominating choice for sequence modeling but it severely suffers from two problems – The first one is its limited ability to capture the long-term dependencies and the second one is the time complexity, which is linear to the sequence length. However, in LocalRNN, these problems are naturally mitigated. Because the LocalRNN is applied to a short sequence within a local window of fixed size, where no long-term dependency is needed to capture. In addition, the computation procedures for processing the short sequences are independent of each other. Therefore, it is very straightforward for the parallel implementation (e.g., using GPUs), which can greatly improve the computation efficiency.

### 3.2 Capturing the Global Long-Term Dependencies with Multi-Head Attention

The RNNs at the lower level introduced in the previous subsection will refine representation of each positions such that it incorporates its local information. In this subsection, we build a sub-layer on top of the LocalRNN to capture the global long-term dependencies. We term it as pooling sub-layer because it functions similarly to the pooling operation in CNNs. Recent works have shown that the multi-head attention mechanism is extremely effective to learn the long-term dependencies, as it allows a direct connection between every pair of positions. More specifically, in the multi-head attention mechanism, each position will attend to all the positions in the past and obtains a set of attention scores that are used to refine its representation. Mathematically, given current representations $h_1, h_2, \cdots, h_t$, the refined new representations $u_t$ are calculated as:

$$u_t = MultiHeadAttention(h_1, h_2, \cdots, h_t) \qquad (4)$$
$$= Concatenation(head_1(h_t), head_2(h_t), \cdots, head_k(h_t))W^o$$

where $head_k(h_t)$ is the result of $k^{th}$ attention pooling and $W^o$ is a linear projection matrix. Considering both efficiency and effectiveness, the scaled dot product is used as the attention function (Vaswani et al., 2017). Specifically, $head_i(h_t)$ is the weighted sum of all value vectors and

the weights are calculated by applying attention function to all the query, key pairs:

$$\{\alpha_1, \alpha_2, \cdots \alpha_n\} = Softmax(\{\frac{<q, k_1>}{\sqrt{(d_k)}}, \frac{<q, k_2>}{\sqrt{(d_k)}}, \cdots, \frac{<q, k_n>}{\sqrt{(d_k)}}\}) \tag{5}$$

$$head_i(h_t) = \sum_{j=1}^{n} \alpha_j v_j$$

where $q$, $k_i$, and $v_i$ are the query, key, and value vectors and $d_k$ is the dimension of $k_i$. Moreover, $q$, $k_i$, and $v_i$ are obtained by projecting the input vectors into query, key and value spaces, respectively (Vaswani et al., 2017). They are formally defined as:

$$q, k_i, v_i = W^q h_t, W^k h_i, W^v h_i \tag{6}$$

where $W^q$, $W^k$ and $W^v$ are the projection matrices and each attention pooling $head_i$ has its own projection matrices. As shown in Eq. (5), each $head_i$ is obtained by letting $h_t$ attending to all the "past" positions, thus any long-term dependencies between $h_t$ and $h_i$ can be captured. In addition, different heads will focus on dependencies in different aspects. After obtaining the refined representation of each position by the multi-head attention mechanism, we add a position-wise fully connected feed-forward network sub-layer, which is applied to each position independently and identically. This feedforward network transforms the features non-linearly and is defined as follows:

$$FeedForward(m_t) = max(0, u_t W_1 + b1)W_2 + b2 \tag{7}$$

Following (Vaswani et al., 2017), We add a residual (He et al., 2016) and layernorm (Ba et al., 2016) connection between all the sub-layers.

### 3.3 OVERALL ARCHITECTURE OF R-TRANSFORMER

With all the aforementioned model components, we can now give a formal description of the overall architecture of an $N$-layer R-Transformer. For the $i^{th}$ layer ($i \in \{1, 2, \cdots N\}$):

$$h_1^i, h_2^i, \cdots, h_T^i = LocalRNN(x_1^i, x_2^i, \cdots, x_T^i) \tag{8}$$

$$\hat{h}_1^i, \hat{h}_2^i, \cdots, \hat{h}_T^i = LayerNorm(h_1^i + x_1^i, h_2^i + x_2^i, \cdots, h_T^i + x_T^i)$$

$$u_1^i, u_2^i, \cdots, u_T^i = MultiHeadAttention(\hat{h}_1^i, \hat{h}_2^i, \cdots, \hat{h}_T^i)$$

$$\hat{u}_1^i, \hat{u}_2^i, \cdots, \hat{u}_T^i = LayerNorm(u_1^i + \hat{h}_1^i, u_2^i + \hat{h}_2^i, \cdots, u_T^i + \hat{h}_T^i)$$

$$m_1^i, m_2^i, \cdots, m_T^i = FeedForward(\hat{u}_1^i, \hat{u}_2^i, \cdots, \hat{u}_T^i)$$

$$x_1^{i+1}, x_2^{i+1}, \cdots, x_T^{i+1} = LayerNorm(m_1^i + \hat{u}_1^i, m_2^i + \hat{u}_2^i, \cdots, m_T^i + \hat{u}_T^i)$$

where $T$ is the length of the input sequence and $x_t^i$ is the input position of the layer $i$ at time step $t$.

**Comparing with TCN:** R-Transformer is partly motivated by the hierarchical structure in TCN Bai et al. (2018), thus, we make a detailed comparison here. In TCN, the locality in sequences in captured by convolution filters. However, the sequential information within each receptive field is ignored by convolution operations. In contrast, the LocalRNN structure in R-Transformer can fully incorporate it by the sequential nature of RNNs. For modeling global long-term dependencies, TCN achieves it with dilated convolutions that operate on nonconsecutive positions. Although such operation leads to larger receptive fields in lower-level layers, it misses considerable amount of information from a large portion of positions in each layer. On the other hand, the multi-head attention pooling in R-Transformer considers every past positions and takes much more information into consideration than TCN.

**Comparing with Transformer:** The proposed R-Transformer and standard Transformer enjoys similar long-term memorization capacities thanks to the multi-head attention mechanism (Vaswani et al., 2017). Nevertheless, two important features distinguish R-Transformer from the standard Transformer. First, R-Transformer explicitly and effectively captures the locality in sequences with the novel LocalRNN structure while standard Transformer models it very vaguely with multi-head attention that operates on all of the positions. Second, R-Transformer does not rely on any position embeddings as Transformer does. In fact, the benefits of simple position embeddings are very

Table 1: MNIST classification task results. Italic numbers denote that the results are directly copied from other papers that have the same settings.

| Model | # of layers / hidden size | Test Accuracy(%) |
|---|---|---|
| RNN (Bai et al., 2018) | - | *21.5* |
| GRU (Bai et al., 2018) | - | *96.2* |
| LSTM (Bai et al., 2018) | *1/ 130* | *87.2* |
| TCN (Bai et al., 2018) | *8 /25* | *99.0* |
| Transformer | 8/32 | 98.2 |
| R-Transformer | 8/32 | 99.1 |

limited (Al-Rfou et al., 2018) and it requires considerable amount of efforts to design effective position embeddings as well as proper ways to incorporate them (Dai et al., 2019). In the next section, we will empirically demonstrate the advantages of R-Transformer over both TCN and the standard Transformer.

## 4 EXPERIMENT

Since the R-Transformer is a general sequential learning framework, we evaluate it with sequential data from various domains including images, audios and natural languages. We mainly compare it with canonical recurrent architectures (Vanilla RNN, GRU, LSTM) and two of the most popular generic sequence models that do not have any recurrent structures, namely, TCN and Transformer. However, since the majority of existing efforts to enhance Transformer are for natural languages, in the natural language evaluation, we also include one recent advanced Transformer, i.e., Transformer-XL. For all the tasks, Transformer and R-Transformer were implemented with Pytorch and the results for canonical recurrent architectures and TCN were directly copied from Bai et al. (2018) as we follow the same experimental settings. In addition, to make the comparison fair, we use the same set of hyperparameters (i.e, hidden size, number of layers, number of heads) for R-Transformer and Transformer. Moreover, unless specified otherwise, for training, all models are trained with same optimizer and learning rate is chosen from the same set of values according to validation performance. In addition, the learning rate annealed such that it is reduced when validation performance reaches plateau.

### 4.1 PIXEL-BY-PIXEL MNIST: SEQUENCE CLASSIFICATION

This task is designed to test model ability to memorize long-term dependencies. It was firstly proposed by Le et al. (2015) and has been used by many previous works (Wisdom et al., 2016; Chang et al., 2017; Zhang et al., 2016; Krueger et al., 2016). Following previous settings, we rescale each $28 \times 28$ image in MNIST dataset LeCun et al. (1998) into a $784 \times 1$ sequence, which will be classified into ten categories (each image corresponds to one of the digits from 0 to 9) by the sequence models. Since the rescaling could make pixels that are connected in the origin images far apart from each other, it requires the sequence models to learn very long-term dependencies to understand the content of each sequence. The dataset is split into training and testing sets as same as the default ones in Pytorch(version 1.0.0) [2]. The model hyperparameters and classification accuracy are reported in Table 1. From the table, it can be observed that firstly, RNNs based methods generally perform worse than others. This is because the input sequences exhibit very long-term dependencies and it is extremely difficult for RNNs to memorize them. On the other hand, methods that build direct connections among positions, i.e., Transformer, TCN, achieve much better results. It is also interesting to see that TCN is slightly better than Transformer, we argue that this is because the standard Transformer cannot model the locality very well. However, our proposed R-Transformer that leverages LocalRNN to incorporate local information, has achieved better performance than TCN.

---

[2]https://pytorch.org

Table 2: Polyphonic music modeling. Italic numbers denote that the results are directly copied from other papers that have the same settings.

| Model | # of layers / hidden size | NLL |
|---|---|---|
| RNN (Bai et al., 2018) | - | *4.05* |
| GRU (Bai et al., 2018) | - | *3.46* |
| LSTM (Bai et al., 2018) | - | *3.29* |
| TCN (Bai et al., 2018) | *4 /150* | *3.07* |
| Transformer | 3/160 | 3.34 |
| R-Transformer | 3/160 | 2.37 |

Table 3: Character-level language modeling. Italic numbers denote that the results are directly copied from other papers that have the same settings.

| Model | # of layers / hidden size | bpc |
|---|---|---|
| RNN (Bai et al., 2018) | - | *1.48* |
| GRU (Bai et al., 2018) | - | *1.37* |
| LSTM (Bai et al., 2018) | *2 /600* | *1.36* |
| TCN (Bai et al., 2018) | *3 /450* | *1.31* |
| Transformer | 3/512 | 1.34 |
| R-Transformer | 3/512 | 1.24 |

## 4.2 NOTTINGHAM: POLYPHONIC MUSIC MODELING

Next, we evaluate R-Transformer on the task of polyphonic music modeling with Nottingham dataset (Boulanger-Lewandowski et al., 2012). This dataset collects British and American folk tunes and has been commonly used in previous works to investigate the model's ability for polyphonic music modeling (Boulanger-Lewandowski et al., 2012; Chung et al., 2014; Bai et al., 2018). Following the same setting in Bai et al. (2018), we split the data into training, validation, and testing sets which contains 694, 173 and 170 tunes, respectively. The learning rate is chosen from $\{5e^{-4}, 5e^{-5}, 5e^{-6}\}$ and dropout with probability of 0.1 is used to avoid overfitting. Moreover, gradient clipping is used during the training process. We choose negative log-likelihood (NLL) as the evaluation metrics and lower value indicates better performance. The experimental results are shown in Table 2. Both LTSM and TCN outperform Transformer in this task. We suspect this is because these music tunes exhibit strong local structures. While Transformer is equipped with multi-head attention mechanism that is effective to capture long-term dependencies, it fails to capture local structures in sequences that could provide strong signals. On the other hand, R-Transformer enhanced by LocalRNN has achieved much better results than Transformer. In addition, it also outperforms TCN by a large margin. This is expected because TCN tends to ignore the sequential information in the local structure, which can play an important role as suggested by (Gehring et al., 2017).

## 4.3 PENNTREEBANK: LANGUAGE MODELING

In this subsection, we further evaluate R-Transformer's ability on both character-level and word-level language modeling tasks. The dataset we use is PennTreebank(PTB) (Marcus et al., 1993) that contains 1 million words and has been extensively used by previous works to investigate sequence models (Chen & Goodman, 1999; Chelba & Jelinek, 2000; Kim et al., 2016; Tran et al., 2016). For character-level language modeling task, the model is required to predict the next character given a context. Following the experimental settings in Bai et al. (2018), we split the dataset into train-

ing, validation and testing sets that contains 5059K, 396K and 446K characters, respectively. For Transformer and R-Transformer, the learning rate is chosen from $\{1, 2, 3\}$ and dropout rate is 0.15. Gradient clipping is also used during the training process. The bpc is used to measure the predicting performance.

For word-level language modeling, the models are required to predict the next word given the contextual words. Similarly, we follow previous works and split PTB into training, validation, and testing sets with 888K, 70K and 79K words, respectively. The vocabulary size of PTB is 10K. As with character-level language modeling, the learning rate is chosen from $\{1, 2, 3\}$ for Transformer and R-Transformer and dropout rate is 0.35. In this task, we also add Transformer-XL (Dai et al., 2019) as one baseline, which has been particularly designed for language modeling tasks and has achieved state-of-the-art performance. Note that to make the comparison fair, we apply the same model configuration, i.e., number of layers, to Transformer-XL. All other settings such as optimizer are the same as its original ones. The learning rate is chosen from $\{0.01, 0.001, 0.0001\}$ and its best validation performance is achieved with $0.001$. Note that, except dropout, no other regularization tricks such as variational dropout and weight dropout are applied. The prediction performance is evaluated with perplexity, the lower value of which denotes better performance.

The experimental results of character-level and word-level language modeling tasks are shown in Table 3 and Table 4, respectively. Several observations can be made from the Table 3. First, Transformer performs only slightly better than RNNs while much worse than other models. The reason for this observation is similar to the case of polyphonic music modeling task that language exhibits strong local structures and standard Transformer can not fully capture them. Second, TCN achieves better results than all of the RNNs, which is attributed to its ability to capture both local structures and long-term dependencies in languages. Notably, for both local structures and long-term dependencies, R-Transformer has more powerful components than TCN, i.e., LocalRNN and Multi-head attention. Therefore, it is not surprising to see that R-Transformer achieves significantly better results. Table 4 presents the results for word-level language modeling. Similar trends are observed, with the only exception that LSTM achieves the best results among all the methods. In addition, the result of Transformer-XL is only slightly better than R-transformer. Considering the fact that Transformer-XL is specifically designed for language modeling and employs the recurrent connection of segments (Dai et al., 2019), this result suggests the limited contribution of engineered positional embeddings.

### 4.4 DISCUSSIONS AND EVALUATION LIMITATIONS

In summary, experimental results have shown that the standard Transformer can achieve better results than RNNs when sequences exhibit very long-term dependencies, i.e., sequential MNIST while its performance can drop dramatically when strong locality exists in sequences, i.e., polyphonic music and language. Meanwhile, TCN is a very strong sequence model that can effectively learn both local structures and long-term dependencies and has very stable performance in different tasks. More importantly, the proposed R-Transformer that combines a lower level LocalRNN and a higher level multi-head attention, outperforms both TCN and Transformer by a large margin consistently in most of the tasks. The experiments are conducted on various sequential learning tasks with datasets from different domains. Moreover, all experimental settings are fair to all baselines. Thus, the observations from the experiments are reliable with the current experimental settings. However, due to the computational limitations, we are currently restricted our evaluation settings to moderate model and dataset sizes. Thus, more evaluations on big models and large datasets can make the results more convincing. We would like to leave this as one future work.

### 5 RELATED WORK

Recurrent Neural Networks including its variants such LSTM (Hochreiter & Schmidhuber, 1997) and GRU (Cho et al., 2014) have long been the default choices for generic sequence modeling. A RNN sequentially processes each position in a sequence and maintains an internal hidden state to compresses information of positions that have been seen. While its design is appealing and it has been successfully applied in various tasks, several problems caused by its recursive structures including low computation efficiency and gradient exploding or vanishing make it ineffective when learning long sequences. Therefore, in recent years, a lot of efforts has been made to develop models

Table 4: Word-level language modeling. Italic numbers denote that the results are directly copied from other papers that have the same settings.

| Model | # of layers / hidden size | Perplexity |
|---|---|---|
| RNN (Bai et al., 2018) | - | *114.50* |
| GRU (Bai et al., 2018) | - | *92.48* |
| LSTM (Bai et al., 2018) | *3 /700* | *78.93* |
| TCN (Bai et al., 2018) | *4 /600* | *88.68* |
| Transformer | 3/128 | 122.37 |
| Transformer-XL | 3/128 | 83.38 |
| R-Transformer | 3/128 | 84.38 |

without recursive structures and they can be roughly divided into two categories depending whether they rely on convolutions operations or not.

The first category includes models that mainly built on convolution operations. For example, van den Oord *et al.* have designed an autoregressive WaveNet that is based on causal filters and dilated convolution to capture both global and local information in raw audios (Van Den Oord et al., 2016). Ghring *et al.* has successfully replace traditional RNN based encoder and decoder with convolutional ones and outperforms LSTM setup in neural machine translation tasks (Gehring et al., 2017; 2016). Moreover, researchers introduced gate mechanism into convolutions structures to model sequential dependencies in languages (Dauphin et al., 2017). Most recently, a generic architecture for sequence modeling, termed as Temporal Convolutional Networks (TCN), that combines components from previous works has been proposed in (Bai et al., 2018). Authors in (Bai et al., 2018) have systematically compared TCN with canonical recurrent networks in a wide range of tasks and TCN is able achieve better performance in most cases. Our R-transformer is motivated by works in this group in a sense that we firstly models local information and then focus on global ones.

The most popular works in second category are those based on multi-head attention mechanism. The multi-head attention mechanism was firstly proposed in Vaswani et al. (2017), where impressive performance in machine translation task has been achieved with Transformer. It was then frequently used in other sequence learning models (Devlin et al., 2018; Dehghani et al., 2018; Dai et al., 2019). The success of multi-head attention largely comes from its ability to learn long-term dependencies through direct connections between any pair of positions. However, it heavily relies on position embeddings that have limited effects and require a fair amount of effort to design effective ones. In addition, our empirical results shown that the local information could easily to be ignored by multi-head attention even with the existence of position embeddings. Unlike previously proposed Transformer-like models, R-Transformer in this work leverages the strength of RNN and is able model the local structures effectively without the need of any position embeddings.

## 6 CONCLUSION

In this paper, we propose a novel generic sequence model that enjoys the advantages of both RNN and the multi-head attention while mitigating their disadvantages. Specifically, it consists of a LocalRNN that learns the local structures without suffering from any of the weaknesses of RNN and a multi-head attention pooling that effectively captures long-term dependencies without any help of position embeddings. In addition, the model can be easily implemented with full parallelization over the positions in a sequence. The empirical results on sequence modeling tasks from a wide range of domains have demonstrated the remarkable advantages of R-Transformer over state-of-the-art non-recurrent sequence models such as TCN and standard Transformer as well as canonical recurrent architectures.

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
