# OpenReview forum: "R-TRANSFORMER: RECURRENT NEURAL NETWORK ENHANCED TRANSFORMER"
_ICLR.cc/2020/Conference — Reject_

### Official Review · AnonReviewer2 · 2019-10-05
**Official Blind Review #2**

**Rating:** 3

**Review:**

Summary: This paper proposes a new architecture, R-Transformer, that blends the Transformer networks and the recurrent networks, so as to better capture both the long- and short-term features. By injecting a local RNN layer at every level of the network, the authors hoped to enhance the Transformer's ability to model locality structure. To demonstrate the modeling power of R-Trasnformer, the paper evaluates the effectiveness of R-Transformer on 4 different sequence tasks (seqMNIST, polyphonic music, character- and word-level PTB).

Contribution: The authors propose an architecture that combines the practices of recurrent and feed-forward sequence models. However, I have major concerns regarding the novelty this paper, the various claims it makes, as well as its experiment setting.

----------------------------------------------------------------------------

Major issues/questions:

1. The techniques proposed by this paper lack novelty. For instance, the entire section 3.2 is simply the original design of the multi-head self-attention by Vaswani et al. The major difference between R-Transformer and the original Transformer is the replacement of positional embedding with an RNN layer, but (in my opinion) the authors did not demonstrate sufficiently its effectiveness via ablative studies (see below). Moreover, some prior works have already exploited the locality structure in Transformers. For instance, [1] showed that a sparse, local Transformer can work extremely well and be very efficient (they achieved SOTA on large-scale char-level language modeling tasks).

2. The experiments do not entirely convince me.
    i) The authors use the "same hidden size for R-Transformer and Transformer." But in fact, as the R-Transformer has one extra RNN/LSTM/GRU layer at every level of the network, the tests were carried out (in effect) using a larger model than the baselines. I think the authors should instead control the # of model parameters, especially since you are running only on small tasks with small-sized models.
    ii) It is nice that the authors tested R-Transformer on a variety of tasks--- this is important. However, in no way do these number achieve the levels of the "state-of-the-art", which the authors claim at the end of Section 1 (e.g., [2] has better number on seqMNIST and character-level PTB, and the Transformer-XL actually achieves <55 perplexity on word-level PTB). Therefore, the numbers don't look particularly appealing to me.
    iii) Lack of more challenging, or large-scale experiments. Sequential MNIST is known to be a relatively simple task, and alternatives such as sequential CIFAR-10 or permuted sequential MNIST are more valuable "small sequence tasks." (e.g., prior works studying long-term sequence model dependencies [3]) I also think that benchmarking R-Transformer on large-scale tasks like WikiText-103 or 1Bword (which prior works like QRNN, TrellisNet, RMC and Transformer-XL all explored) would be much more indicative of its usefulness (and should not be left out in such a paper). For instance, it would be useful to compare on small models with controlled model size, even if larger models that require TPUs are not available.
    iv) Lack of ablative study. Table 4 seems to suggest that Transformer-XL is better than R-Transformer. Is it because of their usage of the relative positional embedding? How does the "finite window size" affect the performance of R-Transformer? Why don't you use Transformer-XL for the other 3 tasks? Lots of interesting questions are unanswered here. More details in (3) below.

3. Regarding the motivation to insert a local RNN at the start of each layer. The authors claim (by citing Al-Rfou et al.) that the positional embedding (PE) have only limited effect. But in fact, Al-Rfou only says that the PE features (which is added only at the first layer, by the design of Vaswani et al.) can get lost once the transformer gets very deep--- which is totally expected. Therefore, Al-Rfou et al. propose to use a learnable embedding for each layer. This in no way suggests that positional embedding has "limited effect" (Note that in their ablative study, when they turned off the learnable PE, they also added back the original PE). Moreover, the authors didn't use convolution to capture local data, because it "completely ignores the sequential information of positions within the local window". That is true. However, **stacking** convolutional layers does capture sequential information. Prior works like [2] showed exactly how temporal convolutions are related to finite-window RNNs. I would suggest the authors to at least compare these different options with ablative studies. I would at least expect a comparison of i) temporal convolution; 2) fixed-window RNN (LocalRNN); 3) unlimited-window RNN; 4) positional embedding; 5) relative PE (which Transformer-XL uses; is that why it's better in Table 4?).

4. The authors claim that the finite-window RNN captures local features. But doesn't that claim only applies to the first layer? Once the first layer multi-head attention mixes all input elements across the sequence, the "local" features fed into the second layer RNN will be, actually, **global** features? Doesn't that "defeat" the purpose of using a local RNN though?

============================

Minor issues that have mild or zero impact on the score:

5. Inconsistent notations. Equation (2) and (3) both describe the LocalRNN(...) function, but clearly have a different input-output signature. Instead of using italics, it's better to have well-defined notations. Another case is Equation (7): you have 'FeedForward(mt)', but 'mt' is not on the right-hand side of the equation at all. Moreover, the symbols used in Eq. (7) are currently inconsistent with Eq. (8).

6. Show the # of parameters in the model in Table 1-4.

7. According to the code released in the dropbox URL, you precompute the index in the finite window and later called 'torch.index_select' to produce a tensor that is 'ksize' larger than the original sequence (cf. Line 130-143 in models/RTransformers.py in the dropbox folder). How does R-Transformer compare to Transformers in terms of speed and memory? Since you convert the batch dimension to (batch_size * seq_len) at every level of the network, I imagine that could slow down the process, especially in high-dimensional/large-scale experiments?

8. There are some typos (e.g., Sec. 4.2, LTSM) and grammatical mistakes in the paper (Sec. 3.3).

9. In Section 4.2, the authors claim that both LSTM and TCN performed better than Transformers on the polyphonic music dataset because "these music tunes exhibit strong local structures". However, the difference between Transformer and LSTM is actually very small, and it's even better than GRU/vanilla RNN. Is that too big a claim to make? In polyphonic music datasets like JSB or Nottingham, there are still longer sequences with longer dependencies...

============================

Overall, I feel that this paper has a good motivation to combine different sequence model families (Transformers, TCNs, RNNs) to improve their modeling power. But at the same time, I feel that the experiments can be a lot stronger, and the paper has limited novelty when compared to prior works. I'm happy to consider adjusting my score if my concerns above are addressed.

[1] "Generating Long Sequences with Sparse Transformers", https://arxiv.org/abs/1904.10509
[2] "Trellis Networks for Sequence Modeling", https://arxiv.org/abs/1810.06682
[3] "Learning Longer-term Dependencies in RNNs with Auxiliary Losses", https://arxiv.org/abs/1803.00144

**Experience Assessment:**

I have published in this field for several years.

**Review Assessment: Checking Correctness Of Derivations And Theory:**

N/A

**Review Assessment: Checking Correctness Of Experiments:**

I carefully checked the experiments.

**Review Assessment: Thoroughness In Paper Reading:**

I read the paper thoroughly.

---

### Official Review · AnonReviewer1 · 2019-10-23
**Official Blind Review #1**

**Rating:** 3

**Review:**

The paper introduces the R-Transformer architecture which adds a local RNN layer before each attention layer in Transformer. The authors claim state-of-the-art performance but only test on tiny tasks where Transformer models have not been heavily optimized and omit the main problem with RNNs - namely their speed. It is an interesting paper still and the locality is a nice way to remedy the speed problem, but the paper lacks a true study and ablations on this main limitation. In summary: the main new idea of the paper is to make RNNs local in Transformer (trying to add RNN layers has been explored before). This idea could be a good tradeoff between full RNN (slow) and no RNN (lack of context), but the following is missing: (1) ablations on speed vs results by locality window, (2) experiments on more widely reported and larger data-sets and models, at least including some language modeling task (wiki or lm1b) and some translation task (like en-de). Without these results, we cannot recommend to accept this paper.

I'm grateful to the authors for their reply. Presently, a very good LM1B or WMT model can be trained for free in Google colab in under a day, so I do not believe it's computationally infeasable to run the experiments I asked for. Even if it took much longer, I'd believe that the time should be invested before acceptance, so I stand by my score.

**Experience Assessment:**

I have published in this field for several years.

**Review Assessment: Checking Correctness Of Derivations And Theory:**

N/A

**Review Assessment: Checking Correctness Of Experiments:**

I carefully checked the experiments.

**Review Assessment: Thoroughness In Paper Reading:**

N/A

---

### Official Review · AnonReviewer3 · 2019-10-23
**Official Blind Review #3**

**Rating:** 6

**Review:**

In this paper, the authors propose a novel transformer model called R-Transformer.
Based on the observation that positional embeddings require a
a lot of design efforts in vanilla Transformer, the authors propose to use local RNNs to
encode local information in replace of positional embeddings.

The paper is well presented and the proposed algorithm is explained in detail.
However, it is not clear how the proposed model obtains positional information of input nodes.
In vanilla Transformer, positional embeddings contains global positional information.
In the R-Transformer, however, the multi-head attention layer will not be able to obtain positional information from local RNN outputs. Will such loss in positional information affect the performance?

The empirical study shows that R-Transformer can outperform vanilla Transformers and recurrent architectures, which is promising. Still, it would be more convincing if the authors could provide comparisons on NMT tasks or larger language modeling datasets such as WikText. Also, I am interested in the training efficiency of these models. How much overhead does the local RNN introduce?

Overall I think this is an interesting paper but experiments could be improved.

**Experience Assessment:**

I have read many papers in this area.

**Review Assessment: Checking Correctness Of Derivations And Theory:**

I carefully checked the derivations and theory.

**Review Assessment: Checking Correctness Of Experiments:**

I assessed the sensibility of the experiments.

**Review Assessment: Thoroughness In Paper Reading:**

I read the paper at least twice and used my best judgement in assessing the paper.

---

### Decision · Program_Chairs · 2019-12-19

**Decision:**

Reject

**Comment:**

The submission proposes a variant of a Transformer architecture that does not use positional embeddings to model local structural patterns but instead adds a recurrent layer before each attention layer to maintain local context. The approach is empirically verified on a number of domains.

The reviewers had concerns with the paper, most notably that the architectural modification is not sufficiently novel or significant to warrant publication, that appropriate ablations and baselines were not done to convincingly show the benefit of the approach, that the speed tradeoff was not adequately discussed, and that the results were not compared to actual SOTA results.

For these reasons, the recommendation is to reject the paper.